# Modifiable Risk Factors and Trends in Changes in Glucose Regulation during the First Three Years Postdelivery: The St Carlos Gestational Diabetes Mellitus Prevention Cohort

**DOI:** 10.3390/nu15234995

**Published:** 2023-12-01

**Authors:** Maria Arnoriaga-Rodriguez, Verónica Melero, Ana Barabash, Johanna Valerio, Laura del Valle, Rocio Martin O’Connor, Paz de Miguel, José A. Diaz, Cristina Familiar, Inmaculada Moraga, Alejandra Duran, Inés Jimenez, Martín Cuesta, María José Torrejon, Mercedes Martinez-Novillo, Isabelle Runkle, Mario Pazos, Miguel A. Rubio, Pilar Matia-Martín, Alfonso L. Calle-Pascual

**Affiliations:** 1Endocrinology and Nutrition Department, Hospital Clínico Universitario San Carlos, Instituto de Investigación Sanitaria del Hospital Clínico San Carlos (IdISSC), 28040 Madrid, Spain; maria.arnoriaga.rodriguez@gmail.com (M.A.-R.); veronica.meleroalvarez10@gmail.com (V.M.); ana.barabash@gmail.com (A.B.); valeriojohanna@gmail.com (J.V.); lauradel_valle@hotmail.com (L.d.V.); rocio@oconnor.es (R.M.O.); pazdemiguel@telefonica.net (P.d.M.); joseangeldiazperez561@gmail.com (J.A.D.); cristinafamiliarcasado@gmail.com (C.F.); inmamgg@hotmail.com (I.M.); aduranrh@hotmail.com (A.D.); i.jimenez.varas@gmail.com (I.J.); martin.cuesta@salud.madrid.org (M.C.); irunkledelavega@gmail.com (I.R.); mario.pazos@salud.madrid.org (M.P.); marubioh@gmail.com (M.A.R.); 2UCM School of Medicine, Medicina II Department, Universidad Complutense de Madrid, 28040 Madrid, Spain; 3Centro de Investigación Biomédica en Red de Diabetes y Enfermedades Metabólicas Asociadas (CIBERDEM), 28029 Madrid, Spain; 4Clinical Laboratory Department, Hospital Clínico Universitario San Carlos, Instituto de Investigación Sanitaria del Hospital Clínico San Carlos (IdISSC), 28040 Madrid, Spain; mariajosefa.torrejon@salud.madrid.org (M.J.T.); mercedes.martineznovillo@salud.madrid.org (M.M.-N.)

**Keywords:** abnormal glucose regulation, changes in glucose regulation, postpartum risk factors

## Abstract

Objective: Evaluation of the influence of potential risk factors (RFs) on glycemic changes at 3 years postpartum. Methods: The glycemic status of 1400 women, in absence of a new pregnancy, was evaluated at 3 months (3 m) and 3 years (3 y) postpartum, after participation in the St. Carlos Gestational Study (2228 normoglycemic pregnant women followed from before gestational week 12 to delivery, from 2015–2017). Abnormal glucose regulation (AGR) was defined as fasting serum glucose ≥ 100 mg/dL and/or HbA1c ≥ 5.7% and/or 2 h 75 g OGTT glucose ≥ 140 mg/dL. In total, 12 modifiable and 3 unmodifiable RFs were analyzed. Results: 3 m postpartum, 110/1400 (7.9%) women had AGR; 3 y postpartum, 137 (9.8%) women exhibited AGR (110 with 3 m normal glucose tolerance [NGT]); 1263 (90.2%) had NGT (83 with 3 m AGR). More women with gestational diabetes mellitus (GDM) progressed to AGR at 3 y (OR: 1.60 [1.33–1.92]) than women without GDM. Yet, most women with 3 m and/or 3 y AGR had no GDM history. Having ≥2 unmodifiable RFs was associated with increased risk for progression to AGR (OR: 1.90 [1.28–2.83]) at 3 y postpartum. Having >5/12 modifiable RFs was associated with increased progression from NGT to AGR (OR: 1.40 [1.00–2.09]) and AGR persistence (OR: 2.57 [1.05–6.31]). Pregestational BMI ≥ 25 kg/m^2^ (OR: 0.59 [0.41–0.85]), postdelivery weight gain (OR: 0.53 [0.29–0.94]), and waist circumference > 89.5 cm (OR: 0.54 [0.36–0.79]) reduced the likelihood of NGT persisting at 3 y. Conclusions: 3-month and/or 3-year postpartum AGR can be detected if sought in women with no prior GDM. Modifiable and unmodifiable RF predictors of AGR at 3 y postpartum were identified. Universal screening for glycemic alterations should be considered in all women following delivery, regardless of prior GDM. These findings could be useful to design personalized strategies in women with risk factors for 3 y AGR.

## 1. Introduction

Gestational diabetes mellitus (GDM) is defined as a state of abnormal glucose tolerance with an onset or initial detection during pregnancy. The prevalence of GDM is on the rise [1], and it represents one of the most frequent medical complications in pregnancy [2]. In addition to its well-known obstetric and neonatal complications [3], GDM is a recognized risk factor (RF) for future maternal and offspring obesity [4], abnormal glucose regulation (AGR), T2DM [4,5], and cardiovascular disease [6,7]. In turn, different RFs have been identified for GDM, including maternal overweight/obesity, delayed childbearing, a prior history of GDM, a family history of T2DM, and ethnicity [1,2,8,9]. Appropriate screening, diagnosis, and treatment [6] are associated with improved pregnancy outcomes. However, there is a lack of consensus regarding long-term follow-up and management in women with a prior history of GDM, as well as in their offspring [2]. 

Scientific societies recommend postnatal assessment of the glycemic status at 3 months postpartum in women diagnosed with GDM [1,6,10,11]. However, abnormal glucose regulation (AGR) 3 months postdelivery has also been observed in women without a history of GDM, suggesting that pregnancy might be a pro-diabetogenic state [12,13,14,15]. In spite of this, risk factors for post-natal AGR have yet to be established, particularly in women exhibiting prior normal glucose regulation (NGR). Yet, both progression to AGR from normoglycemia and reversion to normal glucose regulation (NGR) from AGR can be encountered, as has been described in subjects with prediabetes [16,17]. What is more, detection of glycemic alterations, followed by optimal management, could be crucial for a return to a normoglycemic state. 

The San Carlos GDM Prevention Cohort consists of women participating in a series of studies that have sought to evaluate the effect of a Mediterranean diet (MedDiet), enhanced with extra virgin olive oil (EVOO) and nuts [18,19], on the development of GD, with that diet initiated before the 12th gestational week (GW). Women included in this cohort were evaluated during pregnancy, as well as in short- and long-term follow-up, regardless of prior GDM status. Thus, post-pregnancy evaluation of the study subjects provides a unique opportunity to assess postnatal glycemic changes. It also permits determination of risk factors for the later development of AGR, thereby facilitating identification of potential target populations for the intensification of preventive strategies. 

The aim of this study was to assess rates of reversion to NGR, persistence of AGR, progression to AGR, or persistence of normoglycemia, at 3 years postpartum, as related to the 3-month postpartum glycemic status of the women in the San Carlos GDM cohort. Additionally, risk factors for the aforementioned conditions were sought. 

## 2. Research Design and Methods

### 2.1. Study Design

The study population originated from the San Carlos GDM Prevention Cohort, which includes several studies: 

The San Carlos GDM Prevention Study was a randomized controlled trial (ISRCTN84389045; https://doi.org/10.1186/ISRCTN84389045, accessed on 1 October 2023) directed towards evaluating the effect of an intervention based on a MedDiet enriched with EVOO and nuts on the incidence of GDM. The results indicated the intervention reduced the incidence of GDM and adverse pregnancy outcomes [18]. The recommendations were adopted as standard nutritional management in a real-world study in clinical practice (ISRCTN13389832; https://doi.org/10.1186/ISRCTN13389832, accessed on 1 October 2023) [19]. This latter study was prospective and analyzed a single group of pregnant women receiving a motivational lifestyle interview, with emphasis on daily consumption of EVOO and nuts. Finally, a prospective, randomized intervention study was performed to detect the effect of different components of a MedDiet, in particular EVOO and nuts, on the likelihood of developing GDM in pregnant women with a BMI ≥ 25 and <35 kg/m^2^ (ISRCTN16896947; https://doi.org/10.1186/ISRCTN16896947, accessed on 1 October 2023). 

The Institutional Review Board and the Clinical Ethics Committee of the Hospital Clínico San Carlos approved the aforementioned studies (CI 13/296-E, CI 16/442-E and CI 16/316). Written informed consent forms were provided and signed by all participants.

### 2.2. Study Population

Universal screening for GDM is ordered for all pregnant women in our setting, with a response rate over 95% for attendance for the OGTT between 24–28 GW. Eligible participants were recruited by the Endocrinology and Nutrition Department of the Hospital Clínico San Carlos, a tertiary hospital in Madrid, Spain, with a reference healthcare population of approximately 445,000 patients, from January 2015 to November 2017, and were followed up for a median of 3 years after delivery.

Inclusion criteria were pregnancy in women ≥ 18 years of age, normoglycemia at 8–12 GW (FSG < 92 mg/dL, 5.1 mmol/L), and a single gestation. Exclusion criteria were gestational age at entry ≥ 14 GW, pre-gestational diabetes or a FSG ≥ 92 mg/dL (≥5.1 mmol/L), a multiple pregnancy, intolerance to nuts or EVOO, a new pregnancy during the 3-year follow-up, and medical conditions or pharmacological therapy that could compromise the effect of the intervention and/or the follow-up program.

A total of 3026 normoglycemic women attending their first gestational visit at 8–12 GW were assessed for inclusion. In total, 2529 agreed to participate in the study including postpartum follow-up, and 2228 were followed up until delivery. Of these 2228, 305 were not included in the postpartum study due to a change of address, with another 523 women excluded due to a second pregnancy. Finally, 1400 women (55.4% of the initial cohort) participated in the 3-year postpartum follow-up program, out of 1701 possible candidates (82.3%).

### 2.3. Study Timeline and Intervention

The gestational protocol [18,19] and GDM screening protocol [20] has been previously described.

The postpartum follow-up protocol was as follows:

Visit 1, at 3 months postpartum, consisted of a clinical evaluation, a dietary questionnaire, blood and urine samples, and a motivational lifestyle interview. The latter consisted of dietary recommendations provided by dieticians during 1 h group sessions held with all women participating in the study. Concepts of the MedDiet were reinforced, with emphasis placed on a daily intake of ≥40 mL daily EVOO, and at least 25–30 g of pistachios ≥3 days/week.

Visit 2, at 3 years postpartum, consisted of a clinical evaluation, a dietary questionnaire, blood and urine samples, and a 2 h 75 g OGTT.

### 2.4. Data Collection

Clinical and anthropometric data were collected: maternal age, ethnicity, educational level, employment, smoking status, personal history (hypertension, dyslipidemia, obesity, other diseases), obstetric history (number of pregnancies, prior GDM, miscarriages), and family history (diabetes, hypertension, dyslipidemia, obesity, and MetS, the latter considered when >2 components of MetS were present in at least one first-degree family member). Pre-gestational body weight (BW), gestational BW, height, body mass index (BMI), and blood pressure were registered.

Laboratory tests: Blood samples were drawn after an overnight 8–10 h fast. Fasting serum glucose (FSG) was determined by the glucose oxidase method, serum triglycerides with a colorimetric enzymatic method using glycerol phosphate oxidase p-amino phenazone (GPO-PAP). HbA1c levels were standardized by the International Federation of Clinical Chemistry and Laboratory Medicine using ion-exchange high-performance liquid chromatography in gradient, with a Tosoh G8 analyzer (Tosoh Co., Tokyo, Japan). Serum insulin was determined by a chemiluminescence immunoassay in an Inmmulite 2000 xpi (Siemens, Healthcare Diagnostics, Munich, Germany). The homeostatic model assessment–insulin resistance (HOMA-IR) was calculated as FSG (mmol/L) × fasting serum insulin (FBI) (µU/mL)/22.5. Serum levels of high-density lipoprotein cholesterol (HDL-c) were measured in an Olympus 5800 (Beckman-Coulter, Brea, CA, USA).

Dietary and lifestyle assessment: The Diabetes Nutrition and Complication Trial (DNCT) questionnaire was used to assess physical activity and eating habits, as previously described [21]. The 14-point Mediterranean Diet Adherence Screener (MEDAS) was used to evaluate the degree of adherence to a MedDiet pattern [22]. These were filled out at each visit by a dietician in a personal interview.

### 2.5. Categorization of Glucose Testing at 3 Months and 3 Years Postpartum

Categories based on FSG and/or HbA1c at 3 months after delivery were as follows: women were defined as NGR when exhibiting an FSG < 100 mg/dL (<5.6 mmol/L) and HbA1c < 5.7% (<39 mmol/mol), whereas AGR was diagnosed when FSG was ≥100 mg/dL (5.6 mmol/L) and/or HbA1c ≥ 5.7% (39 mmol/mol). At 3 years postpartum, a 2 h serum glucose following a 75 g OGTT ≥ 140/mg/dL was considered abnormal, whereas a result < 140 was considered normal [23].

### 2.6. Unmodifiable and Modifiable RF

Three non-modifiable RF were evaluated: a family history of T2DM and/or >2 components of the MetS (categorized as 0 if none, 1 if ≥1 presented); parity (0: primiparous, 1: multiparous); age (0: <35, 1: ≥35 years). Women were classified as being in an unfavorable category when at least 2 unmodifiable risk factors coexisted.

Twelve modifiable RFs were evaluated: (i) pre-pregnancy and (ii) 3 month postpartum BMI (0: <25, 1: ≥25 kg/m^2^); (iii) weight change, defined as the difference between pre-pregnancy and 3-month postpartum BW (0: ≤0, 1: >0 kg); (iv) waist circumference as per Spanish criteria (0: <89.5 cm, 1: ≥89.5 cm); (v) hypertension (0: systolic blood pressure (SBP) < 130 and diastolic blood pressure (DBP) < 85 mmHg, 1: SBP ≥ 130 and/or DBP ≥ 85 mmHg); (vi) dyslipidemia (0: HDL-c ≥ 50 mg/dL and triglycerides < 150 mg/dL, 1: HDL-c < 50 mg/dL and/or triglycerides ≥ 150 mg/dL); (vii) post-natal alcohol consumption (0: between 15 and 30 g alcohol/day, 1: <15 or ≥30 g alcohol/day); (viii) smoking habits (0: no or former, 1: smokers). Eating patterns were evaluated using (ix) the nutrition questionnaire (0: ≥4, 1: <4 score) and (x) MEDAS (0: ≥6, 1: <6 score). Physical activity was evaluated with the (xi) activity score (PAS) (0: ≥0, 1: <0 score) and (xii) daily minutes of exercise activity of at least moderate intensity (0: ≥15, 1: <15 min/day). Women were classified in the unfavorable group when >5 modifiable risk factors coexisted.

### 2.7. Study Outcomes

The primary endpoint was to evaluate glycemic status at 3 years postpartum as a function of the 3-month postpartum glycemic state. Specifically, 3-year reversion or persistence of AGR and 3-year progression or persistence of NGR rates were considered.

The secondary endpoint was to identify different pre-gestational, gestational, and 3-month postdelivery risk factors that could influence glycemic changes.

### 2.8. Statistical Analysis

Variables are presented with their number and frequency distribution or the median and interquartile range (IQR). Continuous variables are given as mean and standard deviation (±SD) and were compared using Student’s *t* test or the Mann–Whitney *U* test if the distribution of quantitative variables was not normal, as verified by the Shapiro–Wilk test. Comparison between group characteristics for categorical variables was evaluated by the χ^2^ test.

The magnitude of association between 3-year postpartum glucose regulation status (persistence or reversion of normoglycemia and progression or persistence of abnormal glucose regulation) and modifiable or unmodifiable risk factors was evaluated using the crude odds ratio (OR) and 95% confidence interval (95%CI).

All *p* values are 2-tailed at <0.05. Analyses were performed using SPSS, version 21 (Chicago, IL, USA).

## 3. Results

Women participants were older than non-participants, were more frequently non-smokers, in possession of university degrees, and were qualified workers. Additionally, they showed higher scores in the nutrition and MEDAS questionnaires when compared with non-participants. However, the rates of GDM (20.6% vs. 15.2%), of pre-term deliveries, and of small-for-gestational-age babies was higher in participants. No differences in anthropometric, blood pressure, glycemic, and lipid parameters were found between the two groups (Table 1).

Women with 3-month AGR, and their 3-year evolution

A total of 110 women (7.9%) presented AGR at 3 months postpartum. They were more frequently of Latin American origin and were less likely to possess college degrees than women with NGR. They also had a higher pregestational BW and BMI, and had been diagnosed more often with GDM (44% vs. 19%; *p* < 0.001) as well as preeclampsia. Newborns were more frequently SGA, and the women exhibited a less favorable 3-month postpartum metabolic profile (Table 2). A total of 60% did not have a prior diagnosis of GDM.

Most of these women with 3-month AGR (83/110 (75.5%) presented normoglycemia by the time of the 3-year visit. These women were younger, had a lower BMI, waist circumference, and higher HDL levels at the 3-year visit than the 27/110 (24.5%) in whom AGR persisted. The latter women also had a higher BW than the former and had lost less weight from 3 months to 3 years postdelivery (Table 3).

Women with 3-month NGR, and their 3-year evolution

One thousand two hundred and ninety women were normoglycemic at 3 months postpartum. The majority, 1180/1290 (91.5%), remained so at 3 years (Table 3). However, 110/1290 (8.5%) went from normoglycemia at 3 months to displaying AGR at the 3-year visit. Compared with the former, the latter women were older and displayed a higher BMI and waist circumference and more elevated triglyceride levels at 3 months. Furthermore, pregestational obesity was more prevalent, as was weight gain at both the 3-month and 3-year visits versus pregestational weight, and they had lost less weight between the two postnatal visits compared with those women who remained normoglycemic. The majority of 3-year AGR women had not been diagnosed with GDM.

Women with 3-year AGR

Of 137 (9.8%) women exhibiting AGR 3 years postpartum, 27 (1.9%) also did so at 3 months, whereas the majority, 110 (7.9%), were NGT at 3 months. Of the 1263 (90.2%) women with NGT at 3 years postpartum, 1180 (84.3%) had maintained it, whereas 83 (5.9%) had normalized glucose regulation from 3 months postpartum (Table 3).

Analysis of modifiable and unmodifiable risk factors

Logistic regression analysis was used to identify independent predictors of the glycemic changes. Women with prior GDM were more likely to progress from 3-month normoglycemia to AGR at 3 years postpartum and less likely to remain normoglycemic compared with those who had not developed GDM during pregnancy. The same was the case for women with at least two unmodifiable risk factors. Similarly, pregestational overweight/obesity, the lack of recuperation of pregestational weight at 3 months, and having a central distribution of fat reduced the probability of maintaining NGR from 3 months to 3 years postpartum, thereby increasing the rate of progression from NGR to AGR during this time period.

The presence of >5 unfavorable modifiable risk factors was also associated with a lower probability of maintaining NGR from 3 months to 3 years postpartum (0.74:0.51–0.99), as well as of reversal of 3-month AGR (0.49:0.25–0.97). Thus, these risk factors were associated with an increased risk for progression from NGR at 3 months to AGR after 3 years (1.40:1.00–2.09), as well as with the persistence of AGR (2.57:1.05–6.31, all *p* < 0.05). Data are shown in Table 4.

## 4. Discussion

Gestational diabetes mellitus markedly increases the risk for later development of T2DM [5,24,25,26] and cardiovascular disease [26]. While most studies focus on GDM detection and the risk of developing diabetes later in life, there is limited evidence on the presence or consequences of glycemic dysregulation following a normal gestation.

The current study prospectively evaluated the glycemic status of women at 3 years postpartum, following an initial assessment 3 months postdelivery, regardless of whether they had a prior diagnosis of GDM or not. Three-quarters of the women who had shown AGR at 3 months reverted to NGR after 3 years. However, 8.5% of the women who were normoglycemic at 3 months displayed AGR after 3 years, a finding more frequently observed in women with previous GDM. Yet, over half of the women displaying a deterioration in their postpartum glycemic profile had no prior diagnosis of GDM. Thus, a normal glycemic profile during pregnancy or 3 months postdelivery does not guarantee normoglycemia in young women after pregnancy.

Previous studies deal primarily with follow-up of patients with GDM. Furthermore, they focus on the risk of development of T2DM following delivery [27,28,29]. One study that assessed GDM women at both 3 and 12 months postpartum reported that 17.1% of those with NGT at 3 months progressed to AGR at 12 [30]. A normal 3-month OGTT did not preclude the presence of prediabetes or diabetes 9 months later, detected in 10% of these women [30].

AGR 3 months postpartum was observed in 7.9% of the women we studied. The prevalence of AGR in our population was lower than what has been previously described, with rates ranging from 11 to 36% [31,32,33,34]. However, these studies applied different criteria for the diagnosis of GDM. Nor was there homogeneity in the timing and criteria of postpartum testing. The lower prevalence of AGR shortly after delivery that we observed could also be related to MedDiet dietary habits, and the fact that the women studied were predominantly Caucasian.

The current study broadens the scope of post-pregnancy glycemic evaluation by including women with no prior history of GDM, normally overlooked in studies of postdelivery glucose regulation [31,32,33,34], in spite of the fact that pregnancy is known to deteriorate glucose tolerance [35]. In total, 40% of the women displaying AGR 3 months postpartum in fact had a diagnosis of GDM, while 60%, a majority, did not. These findings suggest that postpartum glycemic screening should be extended to all women following delivery, lest those with undetected AGR be excluded from beneficial clinical interventions.

Women of an older age, as well as those with a family history of type 2 diabetes and/or metabolic syndrome, were less likely to present a normoglycemic status at 3-year follow-up than younger women and those with no such family history. The risk for progression from normoglycemia at 3 months to abnormal glucose regulation at 3 years was higher in women with at least two unmodifiable risk factors. As for modifiable risk factors, a higher pre-gestational and delivery BMI, greater weight gain during follow-up, central obesity, hypertension, and less than 15 min per day of moderate physical activity were associated with a higher probability of progression to abnormal glucose regulation. Furthermore, the combination of >5 modifiable risk factors was associated with the persistence of abnormal glucose regulation.

Some of these aforementioned factors have previously been found to be related to the onset of diabetes later in life in women with prior GDM, including an older age, a waist circumference ≥ 88 cm [28], and a higher BMI [30]. We did not, however, detect independent associations between insulin therapy during pregnancy and the later onset of AGR or dyslipidemia, as other studies have reported [28,29].

More than half (55.4%) of the 2529 women who were studied during pregnancy were included in the 3-month and 3-year follow-up evaluations. They represented over 80% of potential participants, once women who had changed address or had a second pregnancy were excluded. This high response rate could reflect that long-term adherence to a MedDiet enriched with EVOO and nuts, coupled with physical activity, was, in fact, feasible for the women participants. This, in turn, could suggest long-term compliance. The high educational level of the women involved as well as better lifestyle habits could also have been decisive factors in determining participation.

Among the strengths of the current study is the breadth of the assessment of a cohort of women, not limited to patient self-reference but also including both a clinical and laboratory evaluation at the time points studied. Furthermore, to the best of our knowledge, this study provides one of the most comprehensive assessments performed to date of different modifiable and non-modifiable risk factors, not only during pregnancy but also at pre-gestational and early postpartum stages, permitting identification of women prone to abnormal glucose regulation. An additional strength is that early screening at 3 months postpartum was universal for all women, regardless of their risk for glycemic disorders, through follow-up of a MedDiet pattern enhanced with EVOO and nuts. This MedDiet pattern is feasible for all women to follow and could limit the rate of progression to AGR.

The study is not exempt from limitations. First of all, the diagnostic criteria for AGR at 3 months postpartum were based on the use of HbA1c and FSG. The OGTT was not performed at 3 months, but rather at 3 years postpartum. This could underestimate the rate of women with early postpartum AGR. However, >90% of participants were breastfeeding at that time, rendering the OGTT inconvenient for many. In fact, a recent review on the subject of postpartum screening highlights that the OGTT could be a barrier to women’s participation in postdelivery testing [36]. Furthermore, the combination of A1c–FSG appears to detect a similar number of women with prediabetes, while assuring a higher rate of participation. Secondly, alcohol consumption was low in our population. This could be a consequence of the active discouragement of alcohol intake during lactation. Thirdly, participants dedicated little time to exercise, with over half spending merely an average of 15 min daily in physical activity. If the women studied had exercised more, their glycemic results would have presumably improved. However, we cannot rule out that this short amount of time was sufficient to obtain some degree of metabolic benefit. A further limitation is that the use of pharmacological contraception was not registered, and thus, and its influence on glucose regulation cannot be estimated.

In conclusion, we observed that only 40% of the women with AGR in the early postpartum period had a prior diagnosis of GDM. The majority of women displaying this disorder had in fact been normoglycemic throughout pregnancy. Nor had GDM been diagnosed in most of the women displaying 3-year AGR. Thus, all women ought to be candidates for early postpartum screening, with at least an FSG and A1c, and not solely those with previous GDM. Furthermore, additional follow-up should be considered for those women with risk factors for a later appearance of AGR.

Implications for Future Research

Women showing normal glucose regulation shortly after delivery may go on to exhibit glycemic alterations after 3 years. This was seen above all in those over 35 years of age, as well as in women with a family history of metabolic syndrome, pre-pregnancy obesity, those failing to attain their pre-gestation weight, and in women exhibiting a central distribution of body fat. We believe the aforementioned groups should therefore be the object of post-pregnancy monitorization, in an attempt to detect glycemic alterations before the long-term development of T2DM and cardiovascular complications, regardless of their glycemic status at 3 months post-partum. Efforts should be directed towards limiting the number of unfavorable modifiable risk factors in women following pregnancy, whether they have presented GDM during gestation or not, even in the face of a normal glycemic status 3 months post-pregnancy. We recommend that all women at risk for medium-term, 3-year postpartum AGR be the object of specific recommendations and follow-up that should be included in clinical guidelines.

## Figures and Tables

**Table 1 nutrients-15-04995-t001:** Pre-pregnancy and pregnancy characteristics of women included in the San Carlos GDM Prevention Cohort eligible for the postdelivery program.

	Non-Participants (*n* = 1129)	Participants (*n* = 1400)	*p*
Sociodemographic features			
Age (years)	32.1 ± 5.5	33.1 ± 4.9	0.001
Ethnicity			
Caucasian	724 (64.1)	935 (66.8)	
Latin American	364 (32.2)	430 (30.7)	
Others	41 (3.6)	35 (2.5)	0.040
University degree	683 (60.5)	964 (68.9)	0.001
Workers	839 (74.3)	1134 (81.0)	0.001
Smoking status			
Never smokers	604 (53.5)	801 (57.2)	
Current smokers	113 (10.0)	95 (6.8)	0.001
Pregnancies			
First	458 (40.6)	625 (44.6)	
Second	347 (30.7)	415 (29.6)	
Third or above	324 (28.7)	360 (25.7)	0.251
In a previous pregnancy:			
GDM	37 (3.3)	48 (3.4)	0.535
Miscarriage	417 (36.9)	458 (32.7)	0.018
Family history of some components of MetS	463 (41.0)	603 (43.1)	0.130
Pre-pregnancy features			
Body weight (kg)	61.9 ± 11.4	61.7 ± 11.2	0.655
BMI (kg/m^2^)	23.4 ± 4.0	23.4 ± 4.0	0.896
Questionnaires (scores)			
Nutrition	0.1 ± 3.2	0.5 ± 3.1	0.003
MEDAS	4.8 ± 1.7	5.0 ± 1.8	0.016
Physical activity	−1.8 ± 1.0	−1.9 ± 1.0	0.441
Pregnancy and delivery features			
SBP (mmHg) (8–12 GWs)	109 ± 10	109 ± 10	0.167
DBP (mmHg) (8–12 GWs)	67 ± 9	67 ± 9	0.613
FSG (mg/dL) (8–12 GWs)	80 ± 6	80 ± 6	0.560
HbA1 c (%) (8–12 GWs)	5.1 ± 0.3	5.2 ± 0.2	0.181
HOMA-IR (8–12 GWs)	1.2 ± 1.4	1.1 ± 1.3	0.126
Triglycerides (mg/dL) (8–12 GWs)	80 ± 33	82 ± 41	0.901
GDM (24–28 w)	172 (15.2)	289 (20.6)	<0.001
Bodyweight gain (24 GWs)	7.4 ± 4.8	7.1 ± 4.1	0.176
Bodyweight gain (38 GWs)	11.7 ± 6.7	11.8 ± 6.3	0.863
Insulin treatment	40 (23.3)	70 (24.2)	0.432
High BP or preeclampsia	45 (4.0)	53 (3.8)	0.960
Prematurity (<37 GWs)	40 (3.5)	90 (6.4)	0.001
Cesarean section	244 (21.6)	296 (21.2)	0.740
LGA (>90 percentile)	42 (3.7)	50 (3.6)	0.471
SGA (<10 percentile)	34 (3.0)	80 (5.7)	0.001

Data are shown as mean ± standard deviation for quantitative variables or number (%) for qualitative variables. Statistical significance at *p* < 0.05. BMI, body mass index; BP, blood pressure; DBP, diastolic blood pressure; FSG, fasting serum glucose; GDM, gestational diabetes mellitus; HbA1c, glycated hemoglobin; HDL-c, high-density lipoprotein cholesterol; HOMA-IR, homeostatic model assessment–insulin resistance; LGA, large for gestational age, and SGA, small for gestational age (both according to local tables, Hospital Clinic, Barcelona, 2014); MEDAS, Mediterranean Diet Adherence Screener; MetS, metabolic syndrome; SBP, systolic blood pressure; GWs, gestational weeks.

**Table 2 nutrients-15-04995-t002:** Comparison between women with abnormal glucose regulation (AGR) and normal glucose regulation (NGR) at 3 months postdelivery.

	Participants (*n* = 1400)		
	NGR (*n* = 1290)	AGR (*n* = 110)	*p*
Sociodemographic features			
Age (years)	33.1 ± 4.9	33.3 ± 4.6	0.724
Ethnicity			
Caucasian	878 (68.1)	57 (51.8)	
Latin American	383 (29.7)	47 (42.7)	
Others	29 (2.2)	6 (5.5)	0.002
University degree	903 (70.0)	61 (55.5)	0.001
Workers	1051 (81.5)	83 (75.5)	0.231
Smoking status			
Never smokers	734 (56.9)	67 (60.9)	
Current smokers	85 (6.6)	10 (9.1)	0.505
First pregnancy	585 (45.3)	40 (36.4)	0.422
Prior			
GDM	42 (3.3)	6 (5.5)	
Miscarriage	411 (31.9)	47 (42.7)	0.347
Family history MetS (>2 components)	257 (19.9)	25 (22.7)	0.893
Pre-pregnancy features			
Body weight (kg)	61.0 ± 11.0	64.0 ± 14.0	0.012
BMI (kg/m^2^)	23.3 ± 3.9	24.6 ± 4.8	0.001
Questionnaires (scores)			
Nutrition	0.5 ± 3.1	0.1 ± 3.1	0.095
MEDAS	5.0 ± 1.8	4.7 ± 1.7	0.166
Physical activity	−1.9 ± 1.0	−1.8 ± 0.9	0.784
Pregnancy and delivery features			
SBP (mmHg) (8–12 GWs)	108 ± 10	110 ± 11	0.021
DBP (mmHg) (8–12 GWs)	67 ± 9	68 ± 9	0.396
FSG (mg/dL) (8–12 GWs)	80.3 ± 6.0	82.8 ± 5.6	<0.001
HbA1c (%) (8–12 GWs)	5.1 ± 0.2	5.3 ± 0.4	0.045
HOMA-IR (8–12 GWs)	1.1 ± 1.3	1.4 ± 1.5	0.007
Triglycerides (mg/dL) (8–12 GWs)	80 ± 37	104 ± 73	<0.001
GDM (24–28 w)	245 (19.0)	44 (40.0)	<0.001
Bodyweight gain (24 w)	7.2 ± 4.2	6.7 ± 3.7	0.343
Bodyweight gain (38 w)	11.8 ± 6.4	11.3 ± 5.4	0.407
Insulin treatment	55 (4.3)	15 (13.6)	0.204
High BP or preeclampsia	43 (3.3)	10 (9.1)	0.036
Prematurity (<37 w)	79 (6.1)	11 (10.0)	0.088
Cesarean section	267 (20.7)	29 (26.4)	0.567
LGA (>90 percentile)	43 (3.3)	7 (6.4)	0.091
SGA (<10 percentile)	67 (5.2)	13 (11.8)	0.007
3 months postpartum			
Body weight (kg)	66.3 ± 11.6	69.7 ± 12.8	0.015
BMI (kg/m^2^)	25.1 ± 4.3	26.6 ± 4.8	0.005
Weight change (3 m—pregestational)	4.7 ± 5.5	6.0 ± 7.1	0.052
Waist circumference (cm)	85.4 ± 9.5	90.0 ± 9.9	<0.001
SBP (mmHg)	111 ± 12	113 ± 11	0.142
DBP (mmHg)	71 ± 9	72 ± 9	0.583
FSG (mg/dL)	83.6 ± 7.0	94.9 ± 12.1	<0.001
HbA1c (%)	5.2 ± 0.3	5.6 ± 0.3	<0.001
HOMA-IR	1.8 ± 2.4	3.4 ±4.9	<0.001
Triglycerides (mg/dL)	80 ± 43	95 ± 65	0.006
HDL-cholesterol (mg/dL)	64 ± 17	60 ± 13	0.027
Questionnaires (scores)			
Nutrition	3.9 ± 3.5	3.5 ± 3.5	0.071
MEDAS	6.2 ± 1.9	5.9 ± 1.8	0.377
Physical activity	−1.6 ± 0.9	−1.7 ± 0.8	0.618
Exercise activity (min/d)	16 ± 90	3 ± 16	0.065

Data are shown as mean ± standard deviation for quantitative variables or number (%) for qualitative variables. Statistical significance at the *p* < 0.05 level. BMI, body mass index; BP, blood pressure; DBP, diastolic blood pressure; FSG, fasting blood glucose; GDM, gestational diabetes mellitus; HbA1c, glycated hemoglobin; HDL-c, high-density lipoprotein cholesterol; HOMA-IR, homeostatic model assessment–insulin resistance; FSG, fasting serum glucose; LGA, large for gestational age, and SGA, small for gestational age (both according to local tables, Hospital Clinic, Barcelona, 2014); m, months; MEDAS, Mediterranean Diet Adherence Screener; MetS, metabolic syndrome; AGR, abnormal glucose regulation; NGR, normal glucose regulation; SBP, systolic blood pressure; SGA, small for gestational age; W, weeks.

**Table 3 nutrients-15-04995-t003:** Glycemic status change at 3 years (3 y) postdelivery according to metabolic characteristics at 3 months (3 m) postdelivery.

	3 m AGR (110)			3 m NGR (*n* = 1290)		
	3 y NGR Reversion (*n* = 83)	3 y AGR Persistence (*n* = 27)	*p*	3 y AGR Progression (*n* = 110)	3 y NGR Persistence (*n* = 1180)	*p*
Age (years)	32.7 ± 4.6	34.9 ± 4.5	0.031	34.8 ± 4.5	32.9 ± 4.9	0.001
GDM (24–28 GWs)	30 (36.1)	14 (51.2)	0.001	50 (45.5)	195 (16.5)	0.001
Breastfeeding 3 m. n (%)	79 (95)	25 (92)	0.653	101 (92)	1125 (95)	0.165
Exclusive (months)	5.2 + 1.3	4.9 + 1.5	0.121	4.6 + 1.9	4.9 + 1.6	0.095
Mixed (months)	10.4 + 6.9	9.9 + 7.8	0.351	10.1 + 7.4	10.6 + 7.9	0.314
3 m body weight (kg)	69.5 ± 11.8	69.8 ± 12.3	0.008	69.0 ± 11.8	66.0 ± 11.5	0.031
3 m BMI (kg/m^2^)	26.2 ± 4.6	27.6 ± 5.1	0.014	26.3 ± 4.9	25.0 ± 4.2	0.014
<25	57 (68.7)	15 (55.6)		69 (62.7)	859 (72.8)	
25–29.9	16 (19.3)	6 (22.2)		25 (22.7)	232 (19.7)	
≥30	10 (12.0)	6 (22.2)	0.138	16 (14.5)	89 (7.5)	0.034
Pre-gestational obesity	11 (13.3)	3 (11.1)	0.907	13 (11.8)	63 (5.3)	0.005
3 y weight (kg)	64.0 ± 13.4	67.6 ± 10.8	0.023	67.3 ± 11.1	63.0 ± 10.8	0.023
Bodyweight change						
3 m minus pregestational	4.2 ± 6.5	6.6 ± 7.1	0.056	6.6 ± 6.9	4.7 ± 5.5	0.011
3 y minus 3 m	−4.1 ± 3.9	−2.1 ± 5.1	0.026	−0.7 ± 5.0	−2.0 ± 5.6	0.014
3 y minus pregestational	0.8 ± 0.5	0.9 ± 0.6	0.917	4.8 ± 5.6	1.2 ± 2.5	0.033
3 m waist (cm)	88.9 ± 8.9	90.3 ± 10.2	0.036	87.6 ± 8.9	85.2 ± 9.6	0.036
3 m SBP (mmHg)	110 ± 10	113 ± 12	0.071	113 ± 14	110 ± 12	0.071
3 m DBP (mmHg)	68 ± 6	73 ± 9	0.175	73 ± 12	71 ± 9	0.175
3 m FSG (mg/dL)	92 ± 11	96 ± 12	0.021	86 ± 7	83 ± 7	0.001
3 m HbA1c (%)	5.6 ± 0.3	5.7 ± 0.3	0.042	5.3 ± 0.2	5.2 ± 0.3	0.001
3 m HOMA-IR	2.6 ± 2.2	3.5 ± 5.5	0.907	1.8 ± 1.2	1.7 ± 2.4	0.907
3 m HDL-c (mg/dL)	62 ± 13	52 ± 12	0.016	62 ± 19	64 ± 16	0.627
3 m Triglycerides (mg/dL)	90 ± 51	109 ± 94	0.399	82 ± 34	80 ± 43	0.006
3 m Questionnaires (scores)						
Nutrition	3.7 ± 3.6	2.9 ± 3.3	0.429	4.0 ± 3.5	4.1 ± 3.5	0.282
MEDAS	6.1 ± 1.8	5.6 ± 1.6	0.366	6.0 ± 2.1	6.2 ± 1.8	0.377
Physical activity	−1.6 ± 0.9	−1.8 ± 0.6	0.499	−1.5 ± 0.9	−1.6 ± 0.9	0.339
3 m exercise activity (min/d)	32 ± 18	0 ± 6	0.415	14 ± 69	34 ± 96	0.065

Data are shown as mean ± standard deviation for quantitative variables or number (%) for qualitative variables. Statistical significance at the *p* < 0.005 level. AGR, abnormal glucose regulation; NGR, normal glucose regulation; BMI; body mass index; DBP, diastolic blood pressure; FSG, fasting serum glucose; HbA1c, glycated hemoglobin; HDL-c, high-density lipoprotein cholesterol; HOMA-IR, homeostatic model assessment–insulin resistance; FBG, fasting blood glucose; m, months; MEDAS, Mediterranean Diet Adherence Screener; NGR, normal glucose regulation; SBP, systolic blood pressure; w, weeks.

**Table 4 nutrients-15-04995-t004:** Associations between unmodifiable and modifiable risk factors and glycemic status at 3-year follow-up according to metabolic characteristics at 3 months postdelivery.

	Normal Glucose Regulation (NGR)				Abnormal Glucose Regulation (AGR)			
	Persistence (*n* = 1180): NGR at Both 3 Months and 3 Years Postpartum		Reversion (*n* = 83) from AGR at 3 Months to NGR at 3 Years Postpartum		Progression (*n* = 110): NGR at 3 Months, with AGR at 3 Years Postpartum		Persistence (*n* = 27): AGR at Both 3 Months and 3 Years Postpartum	
	*n* (%)	OR (95% CI)	*n* (%)	OR (95% CI)	*n* (%)	OR (95% CI)	*n* (%)	OR (95% CI)
GDM (24–28 w)	195 (78.9)	**0.22(0.15–0.33)**	30 (68.2)	0.51 (0.21–1.23)	50 (20.4)	**1.60 (1.33–1.92)**	14 (31.2)	1.34 (0.88–2.05)
Unmodifiable risk factors								
Family history MetS	225 (19.1)	**0.64 (0.43–0.96)**	21 (25.3)	0.87 (0.70–1.08)	32 (29.1)	**1.63 (1.04–2.54)**	4 (14.8)	0.51 (0.16–1.66)
Parity	634 (53.7)	0.79 (0.55–1.14)	49 (59.0)	0.50 (0.22–1.14)	67 (60.9)	1.29 (0.87–1.93)	21 (77.8)	2.49 (0.89–6.65)
Age ≥ 35 years	577 (48.9)	**0.63 (0.44–0.92)**	41 (49.4)	1.06 (0.86–1.32)	66 (60.0)	**1.64 (1.10–2.46)**	15 (55.6)	1.28 (0.54–3.06)
Unfavorable group ≥ 2	435 (36.9)	**0.56 (0.39–0.80)**	34 (41.0)	0.80 (0.42–1.55)	59 (53.6)	**1.90 (1.28–2.83)**	13 (48.1)	1.34 (0.56–3.20)
Modifiable risk factors								
Preg-BMI ≥ 25 kg/m^2^	307 (26.0)	**0.59 (0.41–0.85)**	27 (32.5)	**0.45 (0.22–0.92)**	43 (39.1)	**1.80 (1.19–2.71)**	14 (51.9)	**2.85 (1.12–7.25)**
Del-BMI ≥ 25 kg/m^2^	321 (27.2)	0.69 (0.48–1.01)	26 (31.1)	0.66 (0.35–1.26)	41 (37.3)	**1.50 (1.00–2.27)**	12 (44.4)	1.75 (0.72–4.27)
Weight change > 0 kg	833 (70.6)	**0.53 (0.29–0.94)**	51 (61.4)	**0.43 (0.20–0.93)**	93 (84.5)	**2.22 (1.10–4.48)**	7 (25.9)	3.64 (0.93–14.39)
WC ≥ 89.5 cm	205 (17.4)	**0.54 (0.36–0.79)**	24 (28.9)	0.76 (0.39–1.48)	32 (29.1)	**2.02 (1.29–3.12)**	10 (37.0)	1.45 (0.58–3.61)
Hypertension	55 (4.7)	0.61 (0.35–1.08)	3 (3.6)	0.84 (0.65–1.08)	12 (10.9)	**1.94 (1.04–4.04)**	6 (22.2)	2.31 (0.35–15.14)
Dyslipidemia	164 (13.9)	0.98 (0.94–1.02)	13 (15.7)	0.64 (0.31–1.29)	12 (10.9)	1.28 (0.72–2.29)	7 (25.9)	1.89 (0.66–5.36)
Alcohol consumption	69 (5.8)	0.99 (0.92–1.05)	2 (2.4)	0.47 (0.17–1.33)	7 (6.4)	0.25 (0.53–2.98)	2 (7.4)	3.24 (0.43–24.20)
Smoking	7 (0.6)	na	0	na	0	na	0	na
Nutrition score < 4	636 (53.9)	0.68 (0.45–1.04)	32 (38.6)	0.79 (0.33–1.85)	71 (64.5)	1.42 (0.97–2.07)	12 (44.4)	1.38 (0.45–4.22)
MEDAS score < 6	683 (57.9)	0.99 (0.95–1.02)	52 (62.7)	0.87 (0.36–2.11)	68 (61.8)	1.19 (0.79–1.66)	16 (59.3)	1.11 (0.57–2.16)
Physical activity < 0	906 (76.8)	0.66 (0.32–1.37)	77 (92.8)	0.57 (0.09–3.58)	99 (90.0)	1.45 (0.76–2.76)	26 (96.3)	2.03 (0.23–17.62)
Exercise activity < 15 min	1026 (86.9)	0.93 (0.84–1.03)	81 (97.6)	na	100 (90.9)	**1.83 (1.00–3.35)**	27 (100)	**1.33 (1.19–1.49)**
Unfavorable group > 5	387 (32.8)	**0.74 (0.51–0.99)**	33 (39.8)	**0.49 (0.25–0.97)**	46 (41.8)	**1.40 (1.00–2.09)**	17 (63.0)	**2.57 (1.05–6.31)**

Odds ratios in bold denote statistical significance at *p* <0.05 level. Each variable in the first column is considered the comparison group vs. not having the variable. Unfavorable group: the presence of two or more unmodifiable risk factors or >5 modifiable risk factors. BMI, body mass index; Del, delivery; MEDAS, Mediterranean Diet Adherence Screener; P-preg, pre-pregnancy; WC, waist circumference.

## Data Availability

The datasets generated during and/or analyzed in the current study are available from the corresponding author upon reasonable request.

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
