# Peer review of "Modifiable Risk Factors and Trends in Changes in Glucose Regulation during the First Three Years Postdelivery: The St Carlos Gestational Diabetes Mellitus Prevention Cohort"

_nutrients, 2023, doi:10.3390/nu15234995_

Round 1
Reviewer 1 Report
Comments and Suggestions for Authors
Dear authors
congratulation what a wonderful research project
Postpartum abnormal glucose levels are often not of interest for the pregnant women and clinicans however they are something very important to pay attention to
in particular keeping in mind how common is GDM
I would like to suggest minor revisions
1) please pay attention to spaces
2) please try to synthetize as much as possible, abstract, introduction and discussion are sometime difficult to follow
3) please add a table in which you summarize the major findings of your research, highlight the Post partum RF that have emerged to associate with higher risk of AG 3 years after
4) please also mention the poor attendence of women to the recomenede post partum OGTT due to a variety of reasons (please read and cite PMID: 3055812)
Author Response
I would like to suggest minor revisions
Thank you very much for your kind and interesting comments
- please pay attention to spaces.
Response: This has been undertaken.
- please try to synthetize as much as possible, abstract, introduction and discussion are sometime difficult to follow.
Response: Large parts of the manuscript have been rewritten to simplify the text.
3) please add a table in which you summarize the major findings of your research, highlight the Post partum RF that have emerged to associate with higher risk of AG 3 years after – proposal
Highlights:
- - Only a few population-based postpartum studies have included modifiable and unmodifiable factors as risk of change in postpartum glycemic status.
- -This study was conducted to assess modifiable and unmodifiable risk factors (RF) in abnormal glucose regulation (AGR) reversion to normal glucose regulation (NGR) and persistence and NGR persistence and progression to AGR during the first three year postpartum including women without GDM during pregnancy
- - At 3-year follow-up, having ≥ 2 unmodifiable RF and having >5/12 modifiable RF were associated with increased progression from NGT to AGR and AGR persistence. Pregestational BMI ≥25 kg/m2), post-delivery weight gain, and waist circumference >89.5 cm reduced the likelihood of NGT persisting at 3-y
- -Universal screening for glycemic alterations should be considered in all women following delivery, regardless of prior GDM. These findings could be useful to design personalized strategies in women with risk factors for 3-y AGR.
Response: Thank you for your comments, suggestions and contributions.
- please also mention the poor attendence of women to the recomenede post partum OGTT due to a variety of reasons (please read and cite PMID: 3055812) –
Response: We have been unable to locate the article you refer to, as the PMID corresponds with: A case of ovarian granulosa cell tumor with invasive adenocarcinoma of the endometrium. A case report and review of the Japanese literatura. S Nakamura 1, N Nakashima, M Ito, M Nagahama, A Nakagawa, A Nakayama, T Koshikawa, J Asai. PMID: 3055812.
However, we will quote a review on the subject.
Response: We have included further information on participation, as follows, in Methods:
A total of 3,026 normoglycemic women attending their first gestational visit at 8-12 GW were assessed for inclusion. Two thousand five hundred twenty-nine agreed to participate in the study including postpartum follow-up, and 2228 were followed-up until delivery. Of these 2228, 305 were not included in the postpartum study due to a change of address, with another 523 women excluded due to a second pregnancy. Finally, 1,400 women (55.4 % of the initial cohort) participated in the 3-year-postpartum follow-up program, out of 1,701 possible candidates (82.3%).

Reviewer 2 Report
Comments and Suggestions for Authors
Interesting, if largely confirmatory, paper.
Major comments:
- Introduction, line 66. "nor have risk factors for subsequent AGR been fully established". There is, in fact, a large literature that the severity of glycemic abnormality during pregnancy is an important determinant of what happens postpartum, as one would expect. In this study, unfortunately, GDM is a categorical variable. Would it be possible to differentiate GDM severity?
- Methods, line 166: a family history of T2DM and/or >2 components of the MetS. I believe this is very hard to establish, since this diagnosis of the MetS entails a 'thorough' clinical exam (abdominal cc, blood pressure) and a fasting blood sample (glucose, HDL-C and TG). How was this done in practice?
- Table 4. One would expect that the "NGR persistence group" (normal glucose metabolism throughout) is set as the 1.00 OR group. So the data presentation is blurry. The terms "progression" and "reversion" are also somewhat enigmatic in this table, you need the text to work it out: it would be better to define the groups by the stage postpartum (3 months vs. 3 years).
- Breastfeeding, and particularly breastfeeding duration, is not included as a modifiable factor. Arguably, breastfeeding is the best prevention against glycemic abnormalities postpartum, since it lessens weight retention and improves insulin resistance. Surprisingly, this crucial factor remains un-examined.
- The same is true for contraception during/after breastfeeding. Studies have clearly shown that some methods (mainly DMPA, the three months injection) can accelerate the development of glycemic abnormalities (see for example Kjos et al). The contraceptive pill may also affect insulin resistance.
Other comments:
- The Abstract is curiously devoid of verbs etc. , possibly to 'save' words. But it reads awkardly: "AGR (was) defined", "137/1400 women presented (with) AGR (at) 3 years postpartum".
- Methods, line 107: "Universal screening for GDM is performed in all pregnant women". I do not believe this, even in the best of circumstances some women book very late during pregnancy (some at delivery), some people refuse glucose testing, etc. You can attain perhaps 95% of the pregnant population, never a 100%: women are not laboratory rats.
- Table 1. For me, the difference between "Caucasian" and "Hispanic" is unclear: Caucasian is Spanish and other European, I presume, and Hispanic Latin American (mestizo?)
- Table 1. Was LGA and SGA truly GA-corrected? On the basis of which published growth charts?
Comments on the Quality of English LanguageAbstract poorly written
Author Response
Thank you very much for your kind and interesting comments
Major comments:
- Introduction, line 66. "nor have risk factors for subsequent AGR been fully established". There is, in fact, a large literature that the severity of glycemic abnormality during pregnancy is an important determinant of what happens postpartum, as one would expect. In this study, unfortunately, GDM is a categorical variable. Would it be possible to differentiate GDM severity?
Response:
The postpartum evaluation in our study includes women both with and without a prior GDM diagnosis in pregnancy. Consider that both Tables 2 and 3 include the GDM rates in each group. When evaluating subgroups, such as women who required insulin treatment, we did not detect differences. Our insulin treatment rates are in fact low, as less than 20% require insulin due to failure of nutritional treatment.
- Methods, line 166: a family history of T2DM and/or >2 components of the MetS. I believe this is very hard to establish, since this diagnosis of the MetS entails a 'thorough' clinical exam (abdominal cc, blood pressure) and a fasting blood sample (glucose, HDL-C and TG). How was this done in practice?
Response:
Women were asked about their family history, if they had first-degree relatives with obesity/overweight, high cholesterol or triglycerides, high blood pressure, blood sugar or diabetes, and if they received any treatment for these conditions. The coincide in a family member of more than 2 components was detected. Obviously, this methodology underestimates the rate of metabolic syndrome published, affecting more than 35% of the Spanish population, but we consider it to be a valid approximation.
- Table 4. One would expect that the "NGR persistence group" (normal glucose metabolism throughout) is set as the 1.00 OR group. So the data presentation is blurry. The terms "progression" and "reversion" are also somewhat enigmatic in this table, you need the text to work it out: it would be better to define the groups by the stage postpartum (3 months vs. 3 years).
- Response: Thank you for your comments. We have inserted at the top of the table the definition of each group, and below the table and what is the reference group that we have considered for each group.
Persistence of NGR means NGR at 3 months and 3 years postpartum. Reversion to NGR means AGR at 3 months and NGR at 3 years postpartum. Progression to AGR means NGR at 3 months and AGR at 3 years postpartum. Persistence of AGR means AGR at 3 months and AGR at 3 years postpartum.
Each variable in the first column is considered the comparison group vs not having the variable. For example, OR (95%CI) of having GDM compared to not having had it, having MetS Family History vs not having it, and so on for the rest of the variables..
Breastfeeding, and particularly breastfeeding duration, is not included as a modifiable factor. Arguably, breastfeeding is the best prevention against glycemic abnormalities postpartum, since it lessens weight retention and improves insulin resistance. Surprisingly, this crucial factor remains un-examined. –
Response:
We have included the breastfeeding variable in table 3. As you can see, there are no statistically significant differences. >90% of participants were breastfeeding > 12 months (mean 5 months exclusive and 10 months more mixed)
- The same is true for contraception during/after breastfeeding. Studies have clearly shown that some methods (mainly DMPA, the three months injection) can accelerate the development of glycemic abnormalities (see for example Kjos et al). The contraceptive pill may also affect insulin resistance. -
Response.
Unfortunately, the use of pharmacological contraceptive methods has not been recorded. We include a comment on limitations of the study.
We have included: A further limitation is that the use of pharmacological contraception was not been registered, and thus, and its influence on glucose regulation cannot be estimated.
Other comments:
- The Abstract is curiously devoid of verbs etc. , possibly to 'save' words. But it reads awkardly: "AGR (was) defined", "137/1400 women presented (with) AGR (at) 3 years postpartum". – new proposal
Response:
The abstract has been rewritten.
- Methods, line 107: "Universal screening for GDM is performed in all pregnant women". I do not believe this, even in the best of circumstances some women book very late during pregnancy (some at delivery), some people refuse glucose testing, etc. You can attain perhaps 95% of the pregnant population, never a 100%: women are not laboratory rats. –
- Response. The response rate is, in fact below 100%, yet over 95%. We have included the following sentence in Methods: Universal screening for GDM is ordered for all pregnant women in our setting, with a response rate over 95% for attendance for the OGTT from 24-28 GW
- Table 1. For me, the difference between "Caucasian" and "Hispanic" is unclear: Caucasian is Spanish and other European, I presume, and Hispanic Latin American (mestizo?)
Response: This corresponds to a different system of identification in Spain and North America. We will adapt to North American criteria, where Hispanic refers to Spanish language speakers, be they European or not. Where we have used Hispanic, we have changed the denomination to Latin American.
- Table 1. Was LGA and SGA truly GA-corrected? On the basis of which published growth charts? –
Response:
LGA and SGA was corrected for gestational age, according to local tables available between 2015-2017, Hospital Clinic, de Barcelona
We insert this comment at the foot of the table : according to local tables, Hospital Clinic, de Barcelona, 2014
Comments on the Quality of English Language
Abstract poorly written
Response
The abstract has been rewritten.

Reviewer 3 Report
Comments and Suggestions for Authors
A commendable effort. To improve readership and scientific interpretation please address the following points-
Line 120- do you have reasons of why women were not followed (3,026- 2,529). Did 1,400 actively consent to the program?
Line 127- definition of control group. It is unclear what the control group since they were educated with the intervention group, was the only difference is that the MedDiet was not "enforced" how did the intervention group adhere to this?
Line 139- Define MetS, assumption it is metabolic syndrome
Line 158- is FSG defined?
Line 190- What normality test was used for distributions? How is the study powered? There is a large difference between NGR and AGR
Line 206- is the percentage reporting prior GDM, this percentage does not match table 1
Line 210: non-participants need to be more clearly defined population. Who are the non-participants, ie did they consent but was not followed up at 3 years? Was this data taken at enrollment? Is this the control group mentioned in methods?
Table 1- unsure how FSG p value is 0.056 when the values are exactly the same
Comments on the Quality of English LanguagePlease use an enligh revision service prior to resubmission. Currently the publication is difficult to follow. Also take care with font and line numbering consistency. Currently font is at different sizes and line numbers are not continuous.
Author Response
A commendable effort. To improve readership and scientific interpretation please address the following points-
Line 120- do you have reasons of why women were not followed (3,026- 2,529). Did 1,400 actively consent to the program?
Response: We have included the information, as follows: A total of 3,026 normoglycemic women attending their first gestational visit at 8-12 GW were assessed for inclusion. Two thousand five hundred twenty-nine agreed to participate in the study including postpartum follow-up, and 2228 were followed-up until delivery. Of these 2228, 305 were not included in the postpartum study due to a change of address, with another 523 women excluded due to a second pregnancy. Finally, 1,400 women (55.4 % of the initial cohort) participated in the 3-year-postpartum follow-up program, out of 1,701 possible candidates (82.3%).
Line 127- definition of control group. It is unclear what the control group since they were educated with the intervention group, was the only difference is that the MedDiet was not "enforced" how did the intervention group adhere to this?
Response: During the postpartum follow-up, the nutritional recommendation was uniform for all women. There was no control group. These nutritional recommendations are based on the Mediterranean diet principles with liberal of the consumption of EVOO and nuts. The reference to a control group refers to prior studies.
A motivational lifestyle interview was used to reinforce dietary concepts. It consisted in guidance from dieticians in a 1-hour group session for all women. These nutritional recommendations are based on the Mediterranean diet principles with liberalization of the consumption of EVOO and nuts , The recommendations comprises the reinforcement of MedDiet adherence and >40 mL daily EVOO consumption (raw and for cooking), and at least 25–30 g of pistachios >3 days/week.
Line 139- Define MetS, assumption it is metabolic syndrome –
Response: It is defined in line 69. the metabolic syndrome (MetS)
Line 158- is FSG defined? –
Response: It is defined in lines 139-140.
Line 190- What normality test was used for distributions? How is the study powered? There is a large difference between NGR and AGR
Response: The Shapiro-Wilk test was applied in statistical analysis, as stated.
Line 206- is the percentage reporting prior GDM, this percentage does not match table 1 –
Response: Prior GDM, refers to the diagnosis prior to the current pregnancy . GDM refers to the gestational diabetes mellitus diagnosis at 24-28 weeks of gestation (15.2% in non-participants and 20.6% in participants) in current gestation (Table 1). To clarify this point, we include in Table 1 and 2 “in a previous pregnancy,” instead of “prior.”
Line 210: non-participants need to be more clearly defined population. Who are the non-participants, ie did they consent but was not followed up at 3 years? Was this data taken at enrollment? Is this the control group mentioned in methods?
Answer. Participants are defined in lines 116-119
Response: Non-participants are those who signed the informed consent at the first gestational visit (8-12 GW) (2529) and were not followed up until delivery or during the 3 years postpartum due to change of address or new pregnancy. All were enrolled at the first gestational visit and signed the informed consent. The reference to control group refers to a previous publication, that studied women during pregnancy.
Table 1- unsure how FSG p value is 0.056 when the values are exactly the same
Response: it's 0.560, and has been corrected. Thank you so very much for pointing this out!!!
Comments on the Quality of English Language
Please use an enligh revision service prior to resubmission. Currently the publication is difficult to follow. Also take care with font and line numbering consistency. Currently font is at different sizes and line numbers are not continuous.
Response: There was been a major revision of the writing by a native English speaker who is also a physician. Differences in font size have been modified.

Round 2
Reviewer 2 Report
Comments and Suggestions for Authors
None